# Energy Metabolism Disturbances in Cell Models of PARK2 CNV Carriers with ADHD

**DOI:** 10.3390/jcm9124092

**Published:** 2020-12-18

**Authors:** Viola Stella Palladino, Andreas G. Chiocchetti, Lukas Frank, Denise Haslinger, Rhiannon McNeill, Franziska Radtke, Andreas Till, Simone Haupt, Oliver Brüstle, Katharina Günther, Frank Edenhofer, Per Hoffmann, Andreas Reif, Sarah Kittel-Schneider

**Affiliations:** 1Department of Psychiatry, Psychotherapy and Psychosomatic Medicine, University Hospital, Goethe University, D-60528 Frankfurt, Germany; viola.stella.palladino@outlook.it (V.S.P.); lukas.ms.frank@googlemail.com (L.F.); andreas.reif@kgu.de (A.R.); 2Department of Child and Adolescent Psychiatry and Psychotherapy, University Hospital Frankfurt, D-60528 Frankfurt, Germany; Andreas.Geburtig-Chiocchetti@kgu.de (A.G.C.); Denise.haslinger@kgu.de (D.H.); 3Department of Psychiatry, Psychotherapy and Psychosomatic Medicine, University Hospital, University of Würzburg, D-97080 Würzburg, Germany; McNeill_R@ukw.de; 4Department of Child and Adolescent Psychiatry, Psychotherapy and Psychosomatic Medicine, University Hospital, University of Würzburg, D-97080 Würzburg, Germany; Radtke_F@ukw.de; 5Institute of Reconstructive Neurobiology, LIFE & BRAIN Center, University of Bonn Medical Faculty & University Hospital Bonn, D-53127 Bonn, Germany; andy.till.0042@gmail.com (A.T.); oliver.bruestle@uni-bonn.de (O.B.); 6LIFE & BRAIN GmbH, Cellomics Unit, D-53127 Bonn, Germany; shaupt@lifeandbrain.com; 7Department 75, Transfer, University of Cologne, 50923 Cologne, Germany; 8Institute of Molecular Biology & CMBI, University of Innsbruck, AT-6020 Innsbruck, Austria; Katharina.Guenther@uibk.ac.at (K.G.); frank.edenhofer@uibk.ac.at (F.E.); 9Spinal Cord Injury and Tissue Regeneration Center Salzburg (SCI-TReCS), Paracelsus Medical University Salzburg, AT-5020 Salzburg, Austria; 10Institute of Humane Genetics, LIFE & BRAIN Center, University of Bonn Medical Faculty & University Hospital Bonn, D-53127 Bonn, Germany; p.hoffmann@uni-bonn.de

**Keywords:** ADHD, hiPSC, PARK2, mitochondria, disease modelling

## Abstract

The main goal of the present study was the identification of cellular phenotypes in attention-deficit-/hyperactivity disorder (ADHD) patient-derived cellular models from carriers of rare copy number variants (CNVs) in the *PARK2* locus that have been previously associated with ADHD. Human-derived fibroblasts (HDF) were cultured and human-induced pluripotent stem cells (hiPSC) were reprogrammed and differentiated into dopaminergic neuronal cells (mDANs). A series of assays in baseline condition and in different stress paradigms (nutrient deprivation, carbonyl cyanide m-chlorophenyl hydrazine (CCCP)) focusing on mitochondrial function and energy metabolism (ATP production, basal oxygen consumption rates, reactive oxygen species (ROS) abundance) were performed and changes in mitochondrial network morphology evaluated. We found changes in *PARK2* CNV deletion and duplication carriers with ADHD in PARK2 gene and protein expression, ATP production and basal oxygen consumption rates compared to healthy and ADHD wildtype control cell lines, partly differing between HDF and mDANs and to some extent enhanced in stress paradigms. The generation of ROS was not influenced by the genotype. Our preliminary work suggests an energy impairment in HDF and mDAN cells of *PARK2* CNV deletion and duplication carriers with ADHD. The energy impairment could be associated with the role of *PARK2* dysregulation in mitochondrial dynamics.

## 1. Introduction

Attention-deficit/hyperactivity disorder (ADHD) is a very heterogeneous disorder, with a broad spectrum of type and severity of symptoms that interfere with personal functioning and negatively impacts social and occupational activities [1,2]. The general population prevalence of ADHD has been described to be between 4–7% in childhood, and 2–4% in the adult population worldwide [3]. The genetic contribution to ADHD has been estimated between 70–80% [4,5], whereas environment is suggested to explain about 22% of ADHD variance [6,7,8,9]. In the past decades, the development of whole-genome scanning methods allowed to clarify the major contribution of copy number variants (CNVs) to genetic variance. CNVs are large, genomic structural variations that comprise deletions, duplications, triplications, and translocations in comparison to a reference genome [10]. Several rare CNV have been associated with ADHD [11,12,13]. There is evidence that the risk for ADHD fits a polygenic liability threshold model. This means that individuals carrying rare large CNVs could develop ADHD by only carrying a lower number of multiple common genetic risk variants [14]. A genome-wide analysis of rare CNVs conducted by Jarick and colleagues specifically detected CNVs in the *PARK2* (=*PRKN*) genetic locus, implicating this target as a candidate gene for ADHD [15]. PARK2 (also known as Parkin/PRKN, a protein with E3 Ubiquitin ligase function), in concert with the Ser/Thr protein kinase PINK1, plays a pivotal role in the regulation of the mitochondria quality control (MQC) system directing processes such as mitophagy (i.e., selective autophagy-mediated degradation of mitochondria), fusion and fission, biogenesis and mitochondrial transport, and is thus involved in the cellular energy balance and oxidative stress response [16]. Different stimuli, both physiological and pathological, can lead to PINK1 accumulation on the mitochondrial outer membrane where it can recruit and activate cytosolic PARK2 by phosphorylation. PARK2 is then able to flag several proteins expressed on the mitochondrial surface and in the cytoplasm with ubiquitin tags, thus marking them for degradation by the ubiquitine-proteasome system or by autophagy [17]. Acting together, PARK2 and PINK1 represent an internal sensor system for disparate cellular homeostasis perturbations [18]. In addition to mitochondria’s well-known role in energy production, they play a pivotal role in general cellular metabolism, intracellular calcium signaling, generation of ROS and stress responses [19]. In the last years, studies have implicated mitochondrial (mt) dysfunction in ADHD in that increased oxidative markers [20], reduced oxygen consumption, and ATP production as well as increased levels of superoxide radicals have been reported [21].

The role of *PARK2* mutations in Parkinson’s patients are relatively well studied and recent studies also utilize human-induced pluripotent stem cell (hiPSC)-derived neuronal cells [22,23] in this context, but no data are available yet regarding the functional consequences of *PARK2* CNVs associated with ADHD in human-derived cellular models.

In this study, we investigated two different cell models; human fibroblast cell lines (HDF) and as well preliminary data from hiPSC-derived midbrain-derived dopaminergic neurons (mDANs). Cells were derived from adult ADHD patients carrying *PARK2* CNVs (deletion and duplication) in comparison to healthy and ADHD wildtype (WT) controls. We conducted experiments focusing on mitochondrial function and energy metabolism. The MQC system is responsible for basal and normal cellular functions but also plays an important role after cellular homeostasis disturbances [18]. Thus, we chose to additionally investigate if baseline genotype differences were more pronounced after a nutrient deprivation paradigm (“starvation”). This displays a form of metabolic cellular stress that has been shown to induce increased *PARK2* expression [24]. We also applied pharmacological treatment using carbonyl cyanide m-chlorophenyl hydrazine (CCCP), an ionophore that depolarizes the mitochondrial membrane thus triggering PINK1 accumulation and subsequent mitochondrial degradation [25]. Finally, we performed a series of assays focusing on mitochondrial function and energy metabolism (ATP production, basal oxygen consumption rates, reactive oxygen species (ROS) abundance).

## 2. Methods

### 2.1. Neuropsychiatric Assessment and Genotyping

Patients were recruited in 2013 at the Department of Psychiatry, Psychosomatic Medicine and Psychotherapy, University Hospital of Würzburg, Germany within a previously published sample [15] (see Appendix A and Appendix A). The healthy controls were recruited among hospital staff and did not report a history of mental disorders, acute or chronic infections, or severe somatic diseases. Only study participants who gave written informed consent were enrolled in the study, which complied with the latest Declaration of Helsinki and was approved by the Ethics Committee of the University of Würzburg (votum no 96/10). Participants were also examined for early signs of Parkinson’s disease by Unified Parkinson’s Disease Rating Scale (UPDRS), Non-Motor Symptom assessment scale for Parkinson’s disease (PD NMS), and the sniffing test to assess olfactory function. Additionally, Substantia nigra volume was assessed by ultrasound and IQ was measured using the MWT-B (Multiple-Choice Vocabulary Intelligence Test, Mehrfachwahl-wortschatz-test, verbal intelligence) [26]. We recruited four ADHD patients and two healthy controls for skin punch biopsies. From three ADHD patients that were known as CNV carriers from the previously published sample [15] and one healthy wildtype control, we further reprogrammed HDF into hiPSC and differentiated them into mDANs (one adult ADHD *PARK2* CNV duplication risk-carrier = PARK2CNV_DUP/ADHD, one adult ADHD *PARK2* CNV deletion risk-carrier = PARK2CNV_DEL_A/ADHD and one healthy CNV non-risk carrier = WT_A/HEALTHY, see Appendix A). From the three additional participants fibroblasts and hiPSC were generated (one adult ADHD PARK2 deletion carrier = PARK2_DEL_B/ADHD, one adult ADHD patient with the wildtype variation = WT/ADHD and another healthy wildtype control = WT_B/HEALTHY) (see Appendix A). For our experiments, we used (i) HDF from all six participants to have a larger number and the generated (ii) mDANs from the described three participants. We obtained venous blood samples from the participants and DNA was isolated from EDTA-monovettes (Sarstedt, Nümbrecht, Germany) by a de-salting method [27]. DNA concentration and quality were assessed by spectrophotometric measurement (Infinite 200 PRO-Tecan, Männedorf, Switzerland). The presence of the *PARK2* risk-CNV was confirmed by Illumina Infinium Omni2.5-8 bead array analysis.

### 2.2. Skin Biopsies and Fibroblast Primary Cultures

Skin biopsies were taken by medically trained personnel (SKS) under local anesthesia (Scandicain, AstraZeneca, Wedel, Germany) using a skin puncher (3 mm^2^) and fibroblast cultures were generated after standard procedure (see Appendix A).

### 2.3. hiPSCs Generation and Pluripotency Assays

CytoTune-IPs 2.0 Sendai Reprogramming Kit (Invitrogen, Carlsbad, CA, USA) was used to reprogram the fibroblasts into hiPSCs following the manufacturer’s protocol (see Supplementary Material). Embryoid body (EB) assay was performed as well as immunostaining and PCR for pluripotency markers in hiPSC and EBs, see Appendix A and Appendix A and Appendix A).

### 2.4. Differentiation of hiPSCs into Neurons with a Midbrain Dopamine Like Phenotype

Dopaminergic differentiation was chosen because *PARK2* genetic mutation are commonly causal for familial Parkinson’s disease and here neurodegeneration in dopaminergic cells plays an important role [28]. On the other hand, the cells in our study were generated from ADHD patients and in ADHD as well dopaminergic dysfunction is suspected [29]. In addition, methylphenidate as the most commonly used ADHD drug is a norepinephrine- and dopamine reuptake inhibitor [30]. The neural differentiation was performed employing a protocol adapted from Kriks et al. [31] as described elsewhere [32]. For basal characterization of dopaminergic markers and dopamine production see Appendix A and Appendix A.

### 2.5. Immunofluorescence and Mitostaining

For immunofluorescence assays cells were grown on Matrigel-coated coverslips. For the analysis of mitochondrial network morphology, fibroblast cells were grown on coverslips and mitochondria were stained using 400 nM MitoTracker^®^ Red CMXRos (ThermoFisher Scientific, Waltham, MA, USA) and imaged with Zeiss Axio Observer.Z1 microscope with ApoTome function (Zeiss, Oberkochen, Germany). Images were analyzed by a semi-automated analysis with Fiji ImageJ software as described elsewhere [33]. Shape descriptors considered were aspect ratio (AR): major_axis/minor_axis and Form factor (FF): (perimeter2/(4π × area)) (details see Appendix A).

### 2.6. Molecular Karyotyping

Genomic DNA from cell lines was extracted with the DNeasy kit (Quiagen, Hilden, Germany) and analyzed on an Illumina Infinium Omni2.5-8 bead array at the Institute of Human Genetics, LIFE&BRAIN, University of Bonn (for further details see Appendix A).

### 2.7. RNA Extraction, Two-Steps Reverse Transcription PCR (RT-PCR), and Quantitative RT-PCR (RT-qPCR)

RNA was isolated using RNeasy-Plus Mini Kit (Qiagen, Hilden, Germany) according to manufacturer’s instructions. RNA quality and absence of gDNA contamination was measured using the Standard Sensitivity RNA Analysis Kit with Fragment Analyzer (Advanced Analytical, Agilent, Santa Clara, CA, USA). Because of the low gene expression of *PARK2* in fibroblasts, samples were pre-amplified before *PARK2* gene expression analysis. Pre-amplification of the target genes was performed with TaqMan PreAmp Master Mix Kit (ThermoFisher Scientific, Waltham, MA, USA) following the manufacturer’s protocol (for predesigned primer and probe sets see Appendix A). Quantitative RT-PCR (RT-qPCR) primers were designed to be intron spanning with NCBI Primer-Blast with the size of the respective intron being greater than 1 kbp (sequence on Appendix A and Appendix A).

### 2.8. Protein Concentration

Protein concentration from whole cell lysate was determined with ADV02 assay (Cytoskeletron Inc, Denver, CO, USA) and absorbance was measured at 600 nm wavelength by the Infinite M200 PRO microplate reader (Tecan, Männedorf, Switzerland). PARK2 protein levels were measured with Human Parkin SimpleStep (Enzyme-linked Immunosorbent Assay) ELISA Kit (Abcam, Cambridge, UK) following the manufacturer’s instructions (for further details see Appendix A).

### 2.9. Stressor Paradigms

For the subsequent experimental procedures, the cell lines were divided into three groups. The “baseline group” was cultured with standard maintenance media; DMEM with 10%FBS for fibroblasts, and complete neurobasal medium for mDANs. The “starvation group” was subjected to a 24-h serum nutrient deprivation paradigm; DMEM-only for fibroblasts, and neurobasal medium without B27 supplementation for mDANs. The “CCCP group” was cultured with standard culturing media supplemented with 10 µM carbonyl cyanide 3-chlorophenylhydrazone (Sigma-Aldrich, Taufkirchen, Germany) for 24 h. For experimental procedures cultured cells from the same line were randomly assigned to one of the treatment groups. Cells were used in similar passage numbers (for further details see Appendix A).

### 2.10. ATP Production, Oxygen Consumption, and ROS Production

All assays were conducted after standard procedure (see Appendix A) using the Plate reader Infinite 200 PRO (Tecan, Männedorf, Switzerland). ATP production was analyzed using the ATPlite Luminescence Assay System (PerkinElmer, Walluf, Germany) according to manufacturer’s instructions. ROS production was measured using the DCFDA/H2DCFDA-Cellular Reactive Oxygen Species Detection Assay Kit (Abcam, Cambridge, GB, USA), according to a protocol suggested by the manufacturer for 24 h of treatment.

### 2.11. Data Analysis

For each experiment we report the single values of each individual replicate that were also used for statistical analysis. Data were analyzed using SPSS (V22, IBM, Armonk, NY, USA). Data were tested for normal distribution by Kolmogorov-Smirnov and Shapiro-Wilk test and parametric or non-parametric tests were applied appropriately. Exploratory one-way ANOVA or two-way ANOVA as well as repeated measures ANOVA (univariate analysis of variance) were used and Kruskal-Wallis test or Mann-Whitney test were performed. If statistical analysis showed significant difference, post-hoc testing was performed (Tukey HD test). Additionally, effect size was calculated (Cohen’s d or Partial eta squares). Correlation between values obtained from HDF and values obtained from mDANs was calculated by bivariate correlation and Pearson’s r reported. The level of significance was set at *p* = 0.05 because of the exploratory approach and we did therefore not correct for multiple testing. For the mDANs, as only *n* = 1 per group was available, only exploratory and descriptive results from the repeated experiments and means of technical replicates are reported. Graphs were plotted using GraphPad Prism 5.01 (GraphPad Software, San Diego, CA, USA). Standard curves were rendered by using CurveExpert 1.4 (https://curveexpert.software.informer.com/1.4/).

## 3. Results

### 3.1. Confirmation of PARK2 CNVs

*PARK2* CNV status was confirmed by an Illumina Infinium Omni2.5-8 bead array that showed *PARK2* CNVs, spanning exon 2 of transcript NM_004562 (Appendix A) (PARK2 CNV hg19 position: PARK2 CNV DUPLICATION/ADHD chr6:162737426-162882874; PARK2 CNV DELETION/ADHD chr6:162719417-162914986).

### 3.2. Neurological and Psychiatric Assessment

None of the clinical tests regarding early signs of Parkinson’s disease was above the clinical threshold in the part III (motor examination) to allow a diagnosis of PD. There were no significant differences between the *PARK2* CNV carriers and the controls (see Appendix A).

### 3.3. Evaluation of Mitochondrial Network Morphology

Given the known physiological role of *PARK2*, we assessed whether the presence of a *PARK2* CNV might exert downstream effects on biological processes regulated by the PARK2-PINK1 interaction [34]. Two main parameters were considered in our study using the HDF. First, the aspect ratio (AR), which mainly is used for a description of the shape of the mitochondria (a value of 1 translates to a circle, numbers > 1 mean a more elongated shape). Second, the form factor (FF), which describes the mitochondrial network branching (high values imply a tubular network and lower values stand for a more fragmented network) [28,35].

Using the mean AR and FF values of the mitochondria present in each HDF (15 cells per cell line/condition) a significant effect of treatment (nutrient deprivation) could be seen for both the indices (AR: F_(1,84)_ = 59.715, *p* ≤ 0.0001, η_p_^2^ = 0.416; FF: F_(1,84)_ = 16.147, *p* ≤ 0.000, η_p_^2^ = 0.246) and the genotype (AR: F_(2,84)_ = 40.067, *p* ≤ 0.0001, η_p_^2^ = 0.488; FF: F_(2,84)_ = 13.713, *p* ≤ 0.000, η_p_^2^ = 0.246), and for the interaction between the two variables (AR: F_(2,84)_ = 35.441, *p* ≤ 0.0001, η_p_^2^ = 0.458; FF: F_(2,84)_ = 22.492, *p* ≤ 0.0001, η_p_^2^ = 0.349). For both AR and FF, the main difference was between *PARK2* CNV duplication/ADHD versus WT/healthy and WT/ADHD control (Tukey HSD *p* ≤ 0.0001) and *PARK2* CNV deletion-carrier/ADHD (Tukey HSD *p* ≤ 0.0001). Our findings suggest that the *PARK2* CNV duplication carrier/ADHD cells with and without nutrient deprivation might exhibit a pronounced elongated mitochondrial shape and tubular branching compared to *PARK2* CNV deletion carrier/ADHD and WTs with and without ADHD (see Figure 1A–C).

### 3.4. PARK2 Gene and Protein Expression

We investigated if there were differences in the *PARK2* gene expression in fibroblasts due to their genotype and/or after nutrient deprivation (starvation) (Appendix A). *PARK2* gene expression was generally low in our HDF cultures which was a finding that differed from previous studies [24] and a pre-amplification step was necessary. *PARK2* gene expression was nominally lower in HDF cells of the *PARK2* deletion carriers in comparison to healthy control and duplication carriers. However, the *PARK2* duplication carriers showed a PARK2 gene expression which also was slightly nominally lower in comparison to wildtype healthy and ADHD controls. Statistical analysis using exploratory ANOVA did not show significant differences in *PARK2* gene expression between *PARK2* deletion and duplication carriers and wildtype controls (with/without ADHD) (Appendix A). Nutrient deprivation (=starvation) led to an increase in *PARK2* gene expression in *PARK2* deletion and duplication carriers, in healthy controls the PARK2 expression remained on the same level after the stress paradigm, however, no significant statistical effect was detected in an exploratory ANOVA analysis (Appendix A).

We further assessed Parkin protein levels both in fibroblasts and dopaminergic neuron lines in baseline conditions, after 24 h of nutrient deprivation and after 24 h treatment with 10 µM carbonyl cyanide 3-chlorophenylhydrazone (CCCP).

Fibroblast cell lines were first explored separately (regarding duplication vs. deletion vs. wildtype) but similar to the *PARK2* gene expression levels, as well the *PARK2* duplication as the *PARK2* deletion carriers showed reduced PARK2 protein levels compared to the wildtype (Figure 2A). An exploratory ANOVA was calculated to assess genotype and treatment effects on PARK2 protein concentration in HDF lines taking the duplication and deletion carriers as one group vs. the wildtype ADHD and healthy control as one group to increase statistical power (Figure 2B). The analysis showed a significant effect of the genotype (F _(2,27)_ = 11.082, *p* ≤ 0.0001, η_p_^2^ = 0.451) and significant effect of treatment (F _(2,27)_ = 13.810, *p* ≤ 0.0001, η_p_^2^ = 0.506) but not a significant interaction between the two fixed factors. Post-hoc comparisons using the Tukey HSD on the treatment effect indicated that the treatment difference was mainly driven by CCCP treatment vs. baseline and vs. starvation (both Tukey HSD *p* ≤ 0.0001) that induced a decrease of PARK2 protein levels. In regard of the genotype, not only fibroblasts lines derived from PARK2 deletion carrier but also PARK2 duplication carrier showed lower levels of PARK2 protein when compared to WT (both Tukey HSD *p* ≤ 0.0001).

The measurement of PARK2 protein levels on differentiated dopaminergic neuronal cells (Figure 2C) revealed a significant interaction between genotype and treatment (F _(4, 9)_ = 4.819, *p* = 0.024, η_p_^2^ = 0.682). The wildtype healthy control mDAN line showed increased PARK2 protein levels in comparison to as well PARK2CNV_DUP/ADHD as PARK2CNV_DEL_A/ADHD at baseline (Figure 2B). Also nutrient deprivation as CCCP treatment decreased PARK2 protein concentration in wildtype mDANs. CCCP treatment led to an increase of PARK2 protein levels in PARK2 duplication carrier mDAN (Figure 2C).

### 3.5. ATP Levels

The total cellular ATP concentration was evaluated under baseline conditions, after 24-h starvation stress and after 24-h treatment with 10 µM CCCP. The HDF cell lines were first explored separately (regarding duplication vs. deletion vs. wildtype) but similar to the other experiments, as well the PARK2 duplication as the PARK2 deletion carriers showed reduced ATP levels/fold changes compared to the wildtype in baseline and after starvation stress. After CCCP treatment the different cell lines showed similar fold changes in comparison to the healthy control at baseline (Figure 3A). In HDF, an exploratory univariate analysis of variance performed on ATP concentration data taking the duplication and deletion carriers as one group vs. the wildtype ADHD and healthy control as one group to increase the statistical power showed a significant difference due to genotype (F_(2,44)_ = 11.462, *p* < 0.0001, η_p_^2^ = 0.343) and treatment (F_(2,44)_ = 43.465, *p* < 0.0001, η_p_^2^ = 0.664), as well as an interaction effect between the two variables (F_(4,44)_ = 4.666, *p* = 0.003, η_p_^2^ = 0.298) (Figure 3C). Both stressor treatments appeared to lower the amount of ATP in comparison to the baseline conditions (both Tukey HSD *p* < 0.0001). Genotype differences were mainly observed between the PARK2 CNV deletion carrier/ADHD and WT/Healthy and WT/ADHD (Tukey HSD *p* < 0.0001), and to a lesser extent between the PARK2 CNV duplication and WT/Healthy and WT/ADHD (Tukey HSD *p* = 0.064). In both cases it appeared that ADHD/PARK2 CNV carrier cells showed lower levels of ATP compared to WT healthy and ADHD controls, both in basal conditions and after 24-h starvation. However, CCCP treatments appeared to affect all genotypes equally (Figure 3A,B).

The mDANs showed lower levels of ATP from cultures subjected to starvation compared to baseline conditions (Tukey HSD *p* < 0.0001, Figure 3B) whereas the CCCP treatment seems to have a lesser effect on mDANs. Genotype was found to have a significant effect on ATP content, and decreased ATP levels were observed in both PARK2 CNV deletion (Tukey HSD *p* < 0.0001) and PARK2 CNV duplication/ADHD compared to WT healthy control (Tukey HSD *p* = 0.004) (see Figure 3C).

### 3.6. Oxygen Consumption Rates (OCR)

We measured the basal extracellular oxygen consumption rate (OCR) in the HDF and mDANs under baseline conditions, 24-h starvation stress, and 24-h CCCP treatment (Figure 4A–F and Figure 5A–C) and additional dimension of energy metabolism. Because the *PARK2* duplication as well as the deletion carriers showed an effect in the same direction, we analyzed the data from the HDF first separately and then together to increase the statistical power. The time course recording was arbitrarily divided into three bins of 30 min each (I_0–30_, I_30–60_, I_60–90_) to avoid multiple testing and because repeated measured ANOVA was technically not possible to conduct with all the measured time points. Student’s tests were run for each interval to determine if there were differences between the genotypes in baseline OCR. Statistical analysis showed significant differences between genotypes both under baseline conditions (Figure 4A,D) and after starvation (Figure 4B,E). HDF of *PARK2* CNVs carriers revealed lower rates of extracellular oxygen consumptions in comparison to WT under baseline conditions (see Figure 4A) and after starvation (Figure 4B,E). We could not detect a significant difference between *PARK2* CNV deletion and duplication carriers and WT healthy and ADHD control after CCCP treatment (see Figure 4C,F). Our data therefore suggest that HDF derived from ADHD patients carrying *PARK2* CNV deletion and duplication display decreased extracellular oxygen consumption compared to WT under both baseline conditions and after starvation stress, but not after CCCP treatment.

To determine if with time the recorded fluorescence would decrease, we calculated an exploratory repeated measures ANOVA on RFU recorded in the intervals. In the baseline conditions, a significant effect of time (F_(2,8)_ = 71.024, *p* < 0.0001) could be seen, but no interaction between time and genotype. This suggests that the fluorescence signal declines over time, possibly because of the utilization of the reagent, which was unaffected by genotype. This finding was similar in the samples subjected to 24-h starvation (F_(2,14)_ = 411,536, *p* < 0.0001), but in this case the decline was higher in healthy and ADHD wildtype control lines, potentially because of the earlier reported elevated OCR compared to the *PARK2* CNV deletion and duplication carriers (F_(2,14)_ = 88.790, *p* < 0.0001).

Because of only having n = 1 per group in the mDANs we could not perform valid statistical tests with biological replicates. Numerically, also in the mDANs, duplication as well as deletion carriers were observed to have a lower OCR under baseline conditions and, although to a lesser degree, after nutrient deprivation (see Figure 5A,B). After CCCP treatment, the mDANs from especially the ADHD *PARK2* deletion carrier showed lower OCR in comparison to the Healthy WT control (see Figure 5C).

### 3.7. Reactive Oxygen Species Production (ROS)

Finally, we analyzed the effect of the different genotypes on the production of cellular ROS in fibroblast (Appendix A). ROS measurement showed a significant effect of the nutrient deprivation and CCCP stress (F_(2,12)_ = 55.208, *p* < 0.0001, η_p_^2^ = 0.902) but not of genotype (F_(1,12)_ = 2.183, *p* = 0.165, η_p_^2^ = 0.154), and also no interaction effect (F_(2,12)_ = 0.768, *p* = 0.485, η_p_^2^ = 0.114). The main differences for treatment were found between baseline and starvation conditions (Tukey HSD *p* < 0.0001), and baseline and CCCP treatment (Tukey HSD *p* < 0.0001). A higher amount of ROS could be observed after starvation (M = 2.441; SD = 0.262) and CCCP treatment (M = 2.268; SD = 0.064) compared to baseline conditions (M = 1.397; SD = 0.184). We did not find a significant difference in ROS response between starvation and CCCP treatment (Tukey HSD *p* = 0.274). In the mDANs, *PARK2* CNV carriers seemed to display a lower amount of ROS in comparison to the healthy WT. Both starvation and CCCP treatment increased the ROS amount in all three mDAN cell lines (Appendix A).

## 4. Discussion

In our study we investigated two patient-derived cellular models, HDF and also preliminary results from hiPSC-derived mDANs, obtained from adult ADHD patients. The patients we derived mDANs from were CNV carriers of deletions or duplications in the *PARK2* gene, which has previously been associated with ADHD and were compared with healthy and ADHD wildtype carriers [15]. We used two stress paradigms, nutrient deprivation and CCCP treatment, to attempt to enhance the cellular pathophenotype in our experiments. The *PARK2*-coded protein parkin is involved in mitochondrial functions such as mitophagy, fusion and fission, biogenesis and mitochondrial transport [36]. Regarding PARK2 gene expression, we could find the lowest levels in the deletion carrier, however, also the duplication carrier had lower PARK2 gene as well as protein concentration in comparison to the wildtypes. The molecular mechanism of how the duplication might lead to decreased PARK2 gene and protein expression needs to be explored in future studies. In HDF, as well nutrient deprivation as CCCP treatment led to decreased PARK2 protein concentrations in all cell lines independent from the genotype. In mDAN this was true for the WT healthy control cell line, but not for the CNV cell lines. However, because of the preliminary and exploratory data on the mDAN, no definite conclusion can be drawn from those findings. We also assessed the morphology of mitochondria in fibroblasts from ADHD *PARK2* CNV carriers in comparison to wildtype carriers. An altered mitochondrial network morphology was observed in the *PARK2* duplication carrier/ADHD. Starvation stress affected all of our fibroblasts independently of genotype and led to a more fragmented mitochondrial network branching with single elements of a rather spherical shape. On a descriptive level, the PARK2 CNV duplication carrier/ADHD under baseline conditions seemed to show a more elongated mitochondrial shape and increased tubular branching compared to *PARK2* CNV deletion carriers/ADHD and WT control and ADHD. Similar alterations were reported in cells derived from Parkinson’s patients carrying mutations in the *PARK2* gene, with both increased tubular branching [28] and fragmented structure [37] reported. Additionally, in high-aged *Park2* knockout mice, more fragmented mitochondria were observed in conjunction with locomotor impairments [38]. Taken together, the data suggests that genetic variants and mutations in *PARK2* might result in protein dysfunction, thereby impacting mitochondrial stability. ADHD and ASD are neurodevelopmental and not neurodegenerative disorders, however, genetic variation in *PARK2* gene and environmental stressor might lead to impairment of mitochondrial structure in the neurodevelopment even though no further neurodegeneration appears later in life.

In the past years, several studies have linked mitochondrial dysfunction to the etiology of several mental disorders [39]. The brain is an organ that is in need of high energy supply, and it is well established that mitochondria play a crucial role in the metabolism of not only energy, but also amino acids, lipids, and steroids; all of which are essential elements for a normally functioning central nervous system [40]. Moreover, specifically in synapses, mitochondria contribute to the maintenance of the membrane potential, facilitate calcium-dependent neurotransmitter release, and activate second messenger pathways [41,42]. After observing altered mitochondrial structure in our HDFs, we investigated mitochondria-related energy metabolism by measuring ATP levels and oxygen consumption rate in HDF and mDANs, under baseline and two stress paradigm conditions. Our findings indicate a significant difference between genotypes in HDF under both baseline and starvation conditions, and a similar result was observed in mDANs which however needs to be evaluated in the future with an independent set of additional cell lines. ADHD *PARK2* CNV duplication and deletion carriers showed lower levels of cellular ATP compared to healthy and ADHD wildtype controls, suggesting a potential dysfunctional MQC system. This may contribute to an aggregation of damaged mitochondria and ATP loss. This finding is supported by a previous study using fibroblast of carriers of different *PARK2* gene mutations. Under baseline conditions, ADHD *PARK2* CNV deletion and duplication carriers also demonstrated decreased extracellular oxygen consumption rates under both baseline and starvation conditions in HDF, which was again reflected in the mDANs. These results replicate previous findings using ADHD cybrid cells; a transgenic cell model that allows the effects of a patient’s mitochondria to be studied in isolation. This study reported decreased levels of ATP production and oxygen consumption, and higher levels of superoxide radicals [21]. CCCP treatment had no effect on ATP levels or oxygen consumption rate in HDF or mDANs, apart from reduced OCR in ADHD *PARK2* CNV deletion carrier mDANs. This finding was unexpected, as CCCP has been shown to cause mitochondrial depolarization in several other studies. However, it could be that the concentrations used in our study were too low [43].

The physiological function of mitochondria leads to the formation of several reactive oxygen species (ROS) and reactive nitrogen species (RNS), which under normal conditions are attenuated by the redox scavenger system [44]. We therefore aimed at evaluating whether the concentration of ROS was different because of genotype and/or stress paradigms. As expected, our data suggested that the cells had increased ROS production under starvation and CCCP treatment compared to baseline. However, there were no genotypic differences regarding ROS levels.

Our results, at least in part, could reflect dysregulated mitophagy (via PARK2/PRKN and PINK1). It has been shown that parkin is transported to the mitochondria which have an impaired energy metabolism, where it ubiquitinates mitofusin (Mfn) to mark them for degradation and mitophagy [45]. If parkin is not functioning properly, impaired mitochondria can accumulate and lead to neurodegeneration [46]. As no neurodegeneration is described in ADHD and ASD, any genetically induced Parkin alterations might have a more subtle effect, which leads to a higher rate of sub-optimal functioning mitochondria. This is supported by our findings of lower ATP content and reduced oxygen consumption rate, but not to an accumulation of defect mitochondria in the cells, at least not in an amount that would cause neurodegeneration. However, we could also show evidence for a stronger vulnerability of the ADHD *PARK2* CNV deletion and duplication carrier cells to nutrient deprivation stress which has also been described in other Parkin-or PINK1-deficient models [43,47].

The regulation of oxidative stress, measured by the level of ROS, was not altered in our ADHD *PARK2* CNV deletion or duplication cells, suggesting there could be alternative compensatory pathways regarding mitochondrial or metabolic stress. Recent studies give evidence of oxidative and nitrosative stress markers being altered in ADHD compared to healthy controls, as reviewed by Lopresti and colleagues [48]. Despite the results of those studies hinting at higher oxidative stress and less sufficient response to oxidative damage both in children and adults with ADHD [49], reported findings are partially contradictory. This is most likely due to inconsistences in markers tested, publication bias, sample collection, and the population studied [48]. However, we investigated ADHD patients that are carriers of a rare genetic variant, so our results cannot be generalized for a majority of ADHD patients.

## 5. Conclusions

Our work suggests that ADHD *PARK2* CNV carriers might have an energy impairment. This impairment could be due to the diverse role of *PARK2* in mitochondrial dynamics, potentially leading to the disruption of normal brain plasticity and cellular resilience [39]. Several highly sensitive temporal windows, such as nervous system development during embryogenesis, could be more susceptible to such perturbations possibly leading to a role in the etiology of neurodevelopmental diseases. During this sensitive developmental stage, mitochondria are involved in the maturation of neural stem cells, proliferation and differentiation, formation of dendritic processes, and synaptic plasticity [50,51,52,53]. PARK2 and PINK1 are the main regulators of the MQC, and thus could play a functional role in psychiatric and neurodevelopmental disorders like subgroups of patients with ADHD. Therefore, as novel therapeutic options, substances that increase mitochondrial functions and decrease oxidative stress like for example antioxidants (e.g., polyphenols) should be investigated in preclinical and clinical studies for their efficacy as (add-on) treatment of ADHD.

### 5.1. Limitations

The results of our study should be considered under several limitations. First, the number of biological repeats (patients and controls) was low, especially in the mDAN, and we did not use isogenic controls. Second, although the models presented in this study are one of the few available cellular models for studying ADHD, *PARK2* CNVs represent a rare variation found just in a small subset of ADHD patients. Furthermore, we did not conduct electrophysiological experiments yet to investigate basal neuronal functions besides the ability to produce dopamine. Additionally, we did not in detail investigate the CNV effect on neuronal function and survival which will be done in future studies.

### 5.2. Data Availability

The data that support these findings are available from the corresponding author, S.K.-S, upon reasonable request. Human subject data will be deidentified to protect confidentiality.

## Figures and Tables

**Figure 1 jcm-09-04092-f001:**
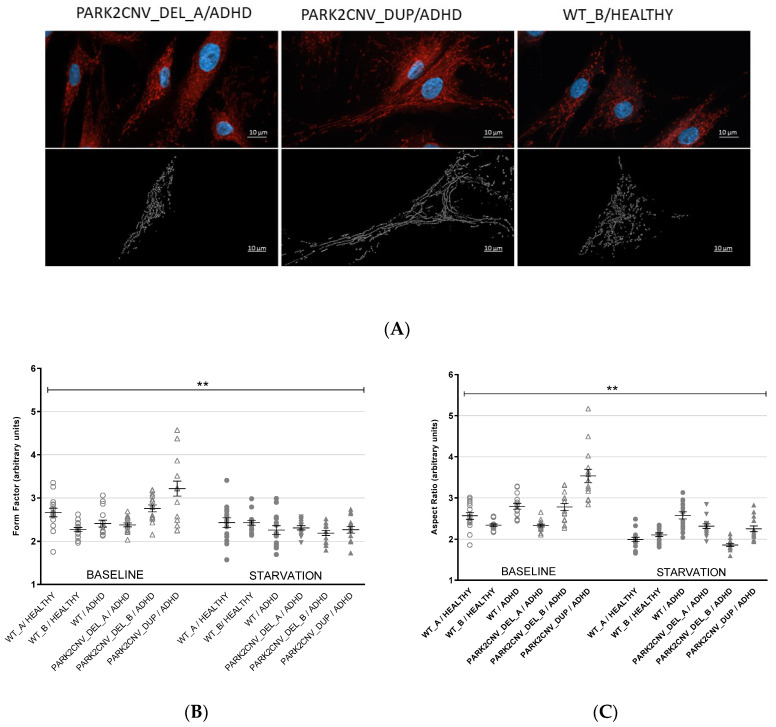
(**A**–**C**): Mitochondrial network analysis. (**A**) Mitochondrial staining (MitoTracker Red CMXRos) in fibroblast cell lines under baseline conditions, and mask of the semi-automated digital image analysis (Fiji/ImageJ) showing different types of network morphology. These images demonstrate a tubular elongated mitochondrial shape (PARK2CNV_DUP/ADHD) and a fragmented round-shaped mitochondrial network (WT_B/HEALTHY and PARK2CNV_DEL_A). Scale bars 10 µm. Graphical representation of the form factor (FF) (1B) and Aspect ratio (AR) (1C). The values were obtained by analyzing 15 cells per line/condition from two independent experiments. The dots represent the mean AR and FF values of all the mitochondria present in each fibroblast cell. In both cases there seem to be a more elongated mitochondrial shape and tubular branching in the PARK duplication carrier (PARK2CNV_DUP/ADHD) compared to the other cells (PARK2CNV_DEL_A/ADHD, PARK2CNV_DEL_B/ADHD, WT_A. WT_B/HEALTHY as well as WT/ADHD). ANOVA was calculated, level of significance was set at *p* = 0.05. ** *p* ≤ 0.01. Data are shown as mean ± SEM.

**Figure 2 jcm-09-04092-f002:**
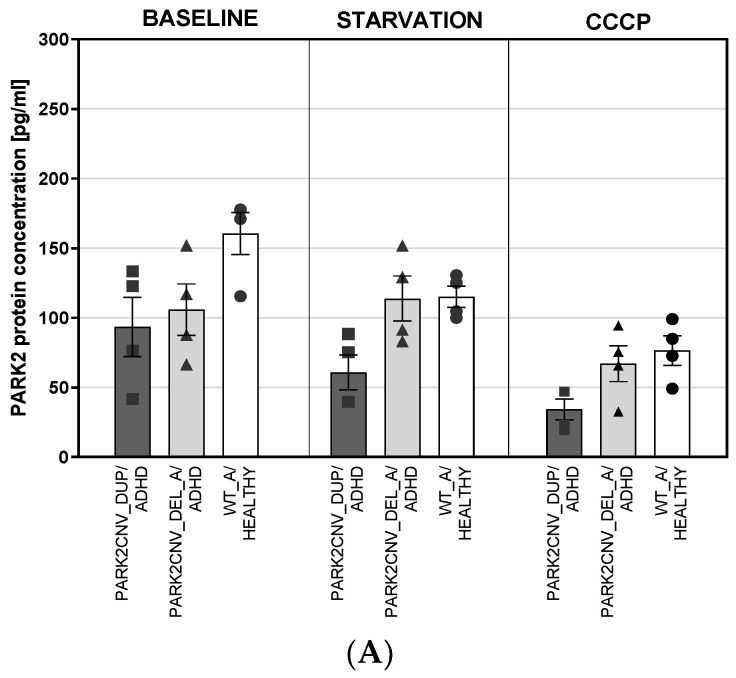
(**A**–**C**): PARK2 protein concentration. PARK2 protein concentration was evaluated in HDF lines (**A** + **B**) and mDANs (**C**) in baseline conditions and after 24 h nutrient deprivation (=starvation) stress and after treatment with 10 µM CCCP for 24 h. PARK2 protein data were obtained from two independent experiments (HDF, only one experiment in mDAN) with samples measured in duplicates. Data are shown as mean ± SEM. Exploratory ANOVA was calculated. **A** show the data from HDF cell lines from *PARK2* CNV duplication vs. deletion vs. wildtype healthy control separately, **C** shows a combined analysis of HDF data of *PARK2* CNV deletion + duplication vs. wildtype ADHD + healthy to increase number of cell lines and statistical power. Level of significance was set at *p* = 0.05. * *p* ≤ 0.05; *** *p* ≤ 0.001.

**Figure 3 jcm-09-04092-f003:**
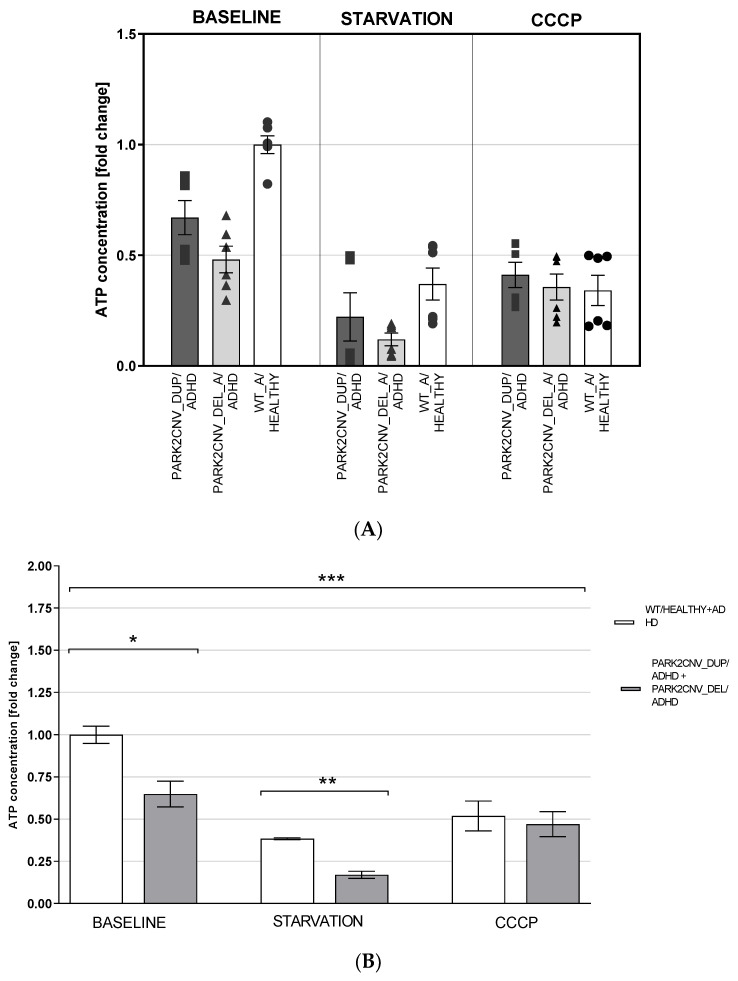
(**A**–**C**): ATP quantification. Cellular ATP content was evaluated in HDF lines (**A**,**B**) and in mDANs (**C**) in baseline conditions, after 24-h starvation stress and after treatment with 10 µM CCCP for 24 h. Fold difference was calculated against WT in baseline conditions. Data were obtained from two independent experiments with samples measured in triplicate. Data are shown as mean ± SEM. Exploratory ANOVA was calculated. Figure A shows the data of HDF cell lines from *PARK2* CNV duplication vs. deletion vs. wildtype healthy control separately, Figure B shows a combined analysis of *PARK2* CNV deletion + duplication vs. wildtype ADHD + healthy to increase number of HDF cell lines and statistical power. Level of significance was set at *p* = 0.05. * *p* ≤ 0.05; ** *p* ≤ 0.01; *** *p* ≤ 0.001.

**Figure 4 jcm-09-04092-f004:**
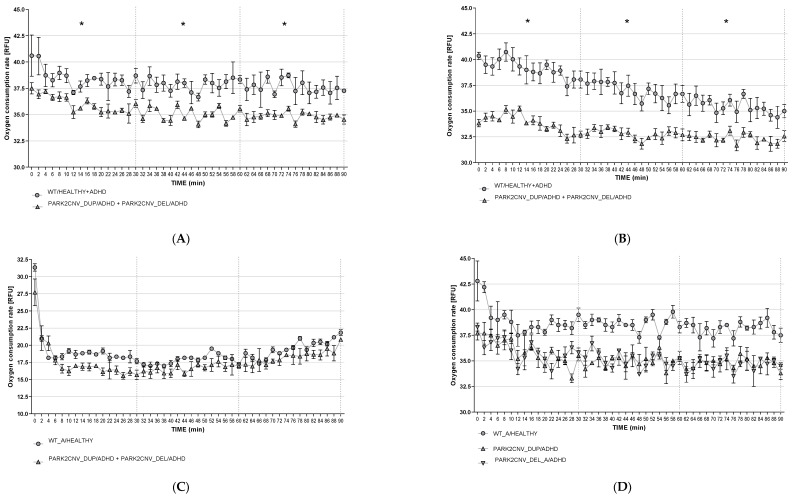
(**A**–**F**): Oxygen consumption rate in human-derived fibroblasts (HDF). Basal oxygen consumption was measured under baseline (**A**,**D**), starvation (**B**,**E**), and CCCP treatment conditions (**C**,**F**). Fluorescence (RFU-*y*-axis) signal correlating with the respiration was recorded every two minutes for 90 min. Data are presented as mean of two independent experiments with samples measured in triplicate. Figure 4A–C show a combined analysis of *PARK2* CNV deletion + duplication vs. wildtype healthy+ADHD wildtype to increase number of HDF cell lines and statistical power). HDF derived from ADHD patients carrying *PARK2* CNVs showed lower rates of extracellular oxygen consumptions compared to WT in all the intervals both under baseline conditions (I_0–30_: CNV carriers (M = 36.056; SD = 0.106), WT (M = 38.493; SD = 1.276); I_30–60_: CNV carriers (M = 35.026; SD = 0.068), WT (M = 37.859; SD = 1.209); I_60–90_: CNV carriers (M = 34.841; SD = 0.293), WT (M = 37.626; SD = 1.279) (see Figure 4A) and after starvation (I_0–30_: CNV carriers (M = 34.938; SD = 1.267 ), WT (M = 39.160; SD = 1.398); I_30–60_: CNV carriers (M = 33.907; SD = 1.296), WT (M = 36.970; SD = 1.527); I_60–90_: CNV carriers (M = 33.593; SD = 1.427), WT (M = 35.430; SD = 1.265). Figure 4D–F shows the data of HDF cell lines from *PARK2* CNV duplication vs. deletion vs. wildtype healthy control separately. There was a significant difference between the genotypes in all intervals in baseline: I_0–30_: t_(4)_ = 3.292, *p* = 0.030, *d* = 2.692, *r* = 0.803; I_30–60_: t_(4)_ = 4.054, *p* = 0.015, *d* = 3.310, *r* = 0.856; I_60__–__90_: t_(4)_ = 3.675, *p* = 0.0321, *d* = 3.001, *r* = 0.832 and as well after starvation: starvation I_0__–__30_: t_(4)_ = 6.224, *p* = 0.003, *d* = 5.082, *r* = 0.931; I_30__–__60_: t_(4)_ = 4.351, *p* = 0.012, *d* = 3.553, *r* = 0.871; I_60__–__90_:t_(4)_ = 3.842, *p* = 0.018, *d* = 3.337, *r* = 0.843. Data are shown as ± standard error of mean (SEM). Exploratory repeated measure ANOVAs and t-tests were calculated. Level of significance was set at *p* = 0.05. * *p* ≤ 0.05.

**Figure 5 jcm-09-04092-f005:**
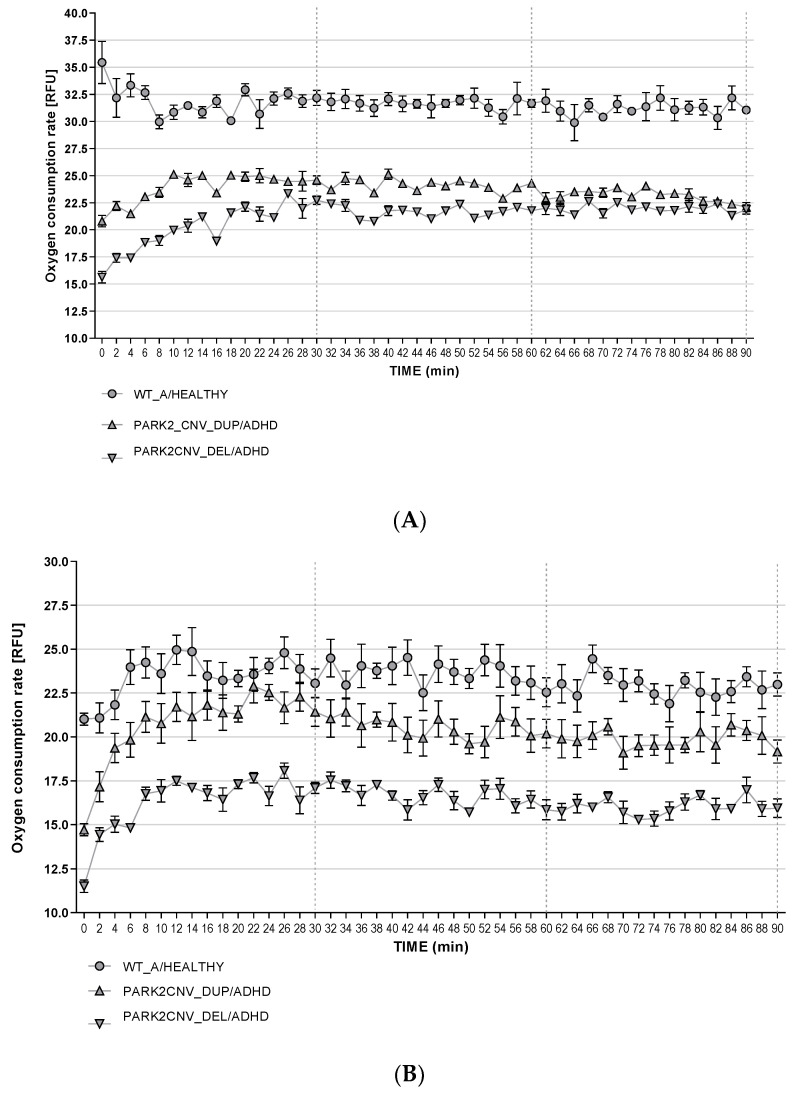
(**A**–**C**): Oxygen consumption rate in mDAN. Basal oxygen consumption rate was evaluated both under baseline (**A**), starvation (**B**), and CCCP treatment conditions (**C**). Fluorescence (RFU-*y*-axis) signal correlating with the respiration was recorded every two minutes for 90 min. Data are presented as mean of two independent experiments with samples measured in triplicates. Because the biological replicates were n = 1 for each group, the data shown here are only descriptive.

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
