# Peer review of "Energy Metabolism Disturbances in Cell Models of PARK2 CNV Carriers with ADHD"

_jcm, 2020, doi:10.3390/jcm9124092_

Round 1
Reviewer 1 Report
Thank you for giving me the opportunity to review this excellent article on energy metabolism disturbances in cell models of PARK2CNV carriers of ADHD.
It uses the cutting edge technology of induced pluripotent stem cells in patients with ADHD that have a PARK2 gene mutation. They access the mitochondrial functioning and energy metabolism by evaluating different parameters such as oxidative stress (and other stress related paradigms) in dopaminergic neurone derived from the patients induced pluripotent stem cells.
They are elucidating the possible role of that genetic mutation on the development of neurodevelopment disorders considering key pathways such as oxidative stress. This paper is a very interesting body of work contributing to solving the riddle of the aetiology of developmental disorders. Due to its complex and cutting edge methodology it adds important knowledge. The researchers team is known for their expertise in ADHD and cellular models such as stem cell methods. All in all, I would most definitively highly recommend this article for publication in your journal, there are just minor language/typing errors that need revision such as listed bellow, but other than that it is excellent work and I can only emphasise that I strongly recommend it for publication.
typos: line 43: not influenced by the genotype. line 117 it should be Nuernberg instead of Nuermbrecht,
Author Response
We thank the reviewer for the praise of our work. We have corrected the typos in the revised version of the manuscript. But the headquarters of the company Sarstedt is acutally really in a town called Nümbrecht, it is not Nuernberg, see here:
https://www.sarstedt.com/fileadmin/user_upload/99_Broschueren/Broschueren_neu_01.2016/653a_image_Brochure_US_US_neu.pdf
Reviewer 2 Report
In this manuscript, the authors investigate the metabolic state in ADHD patients derived cells and the role of copy number variants. The subject is interesting and the methods heterogeneos, even if conducted in as small number of subjects.
Here are my suggestions for a clearer presentation of the results:
- please improve the clarity of the figures: the legends are too small to be read. I suggest to adopt abbreviatons to indicate the subjects and just clearly ditinguish healthy and ADHD subjects. As the figures themselves may be too small, either select the ost relevant ones or separate the panels in different figures
- please better identify the potential clinical/theraputical meaning of the results. For therapy or prevention?
- please add a list of abbreviations
Kind regards
Author Response
Firstly, we thank the reviewer for the positive evaluation of our preliminary study.
The reviewer is right about the small figures, we are presenting the figures in the revised version of the manuscript as single figures in a larger size for better readability. Therefore it should be much easier now to distinguish the different genotypes and ADHD and control lines.
We additionally have added a paragraph to the conclusion section answering the reviewers question after clinical implications of our preliminary results as follows:
“Therefore, as novel therapeutic options, substances that increase mitochondrial functions and decrease oxidative stress like for example antioxidants (e.g. polyphenols) should be investigated in preclinical and clinical studies for their efficacy as (add-on) treatment of ADHD.”
Furthermore, we have added a abbreviation list in the revised manuscript.